# A soccer-based intervention improves incarcerated individuals' behaviour and public acceptance through group bonding

Martha Newson ●[1]✉, Linus Peitz ●[1,2], Jack Cunliffe ●[3] & Harvey Whitehouse ●[1]

As incarceration rates rise globally, the need to reduce re-offending grows increasingly urgent. We investigate whether positive group bonds can improve behaviours among incarcerated people via a unique soccer-based prison intervention, the Twinning Project. We analyse effects of participation compared to a control group (study 1, $n = 676$, $n = 1,874$ control cases) and longitudinal patterns of social cohesion underlying these effects (study 2, $n = 388$) in the United Kingdom. We also explore desistance from crime after release (study 3, $n = 249$) in the United Kingdom and the United States. As law-abiding behaviour also requires a supportive receiving community, we assessed factors influencing willingness to employ formerly incarcerated people in online samples in the United Kingdom and the United States (studies 4–9, $n = 1,797$). Results indicate that social bonding relates to both improved behaviour within prison and increased willingness of receiving communities to support re-integration efforts. Harnessing the power of group identities both within prison and receiving communities can help to address the global incarceration crisis.

If the purpose of prison is to reduce crime by serving as a deterrent and pathway to reform, it would seem to be ineffective in most countries around the world[1–3]. Indeed, imprisonment may have the opposite effect, providing opportunities for incarcerated people to form new relationships and habits of thinking and behaviour that increase the risk of recidivism following release[4,5]. The costs of re-offending worldwide are vast, both in terms of the tax burdens associated with incarceration and the sufferings of victims; for example, the costs of re-offending exceed £18 billion in the United Kingdom and US$5 trillion in the United States every year[6,7].

Although there are many reasons for high rates of re-offending and stubbornly large prison populations, including problematic penal systems, two particularly prevalent risk factors may be the adoption of criminal (rather than law-abiding) attitudes or values and the reluctance of receiving communities to re-integrate formerly incarcerated individuals[8]. Re-integration challenges include assisting formerly incarcerated people to find suitable housing and employment

opportunities. Importantly, re-integration may be particularly problematic for formerly incarcerated people with intersectional identities; for example, in many countries ethnic minorities with criminal records may face particularly harrowing obstacles to securing jobs[9,10].

Both prison- and community-based impediments to 'going straight' are at least partly rooted in group psychology. The formerly incarcerated may cleave to groups and identities that foster criminality in the absence of alternative support networks[11], while society at large tends to stigmatize and exclude them despite socio-economic advantages to successfully re-integrating formerly incarcerated people into the labour market[12]. Here we consider the potential for strong forms of group alignment to change this situation by enabling people in prison to bond with law-abiding groups. Such interventions can also, in the right circumstances, encourage the general public to welcome the formerly incarcerated back into the community by providing a platform through which employers are able to bond with formerly incarcerated people.

[1]Centre for the Study of Social Cohesion, University of Oxford, Oxford, UK. [2]Institute for Lifecourse Development, University of Greenwich, London, UK. [3]School for Social Policy, Sociology and Social Research, University of Kent, Canterbury, UK. ✉e-mail: m.newson@greenwich.ac.uk

Sport may be the ideal platform to capture this reciprocal dynamic. However, a recent meta-analysis suggested that while sports programmes in prison have a significant, moderate effect on behaviour, sounder evaluative designs are required to understand why this is the case[13], with high-profile calls for more thorough investigations into the processes by which sports interventions work[14]. Such programmes could include anything from running[15] to soccer[16]. We focus in this paper on the effects of participation in the Twinning Project, which pairs over 70 major soccer clubs to their local prison to provide coaching skills to people in prison[17]. Initially launching in England and Wales, the Twinning Project now runs in four continents with sites in the United States, Italy, Australia and South Africa. These courses focus not only on the development of coaching techniques but also on transferable skills such as relationship building, self-control and healthy living. We tackle this from a social identity perspective: to the extent that the intervention provides people in prison with an opportunity to bond to the Twinning Project, the aim is to attach them to law-abiding values and communities. Further, because soccer also inspires strong forms of support among the wider public in the United Kingdom[18], the aim is to leverage the cohesion associated with soccer fandom to further foster re-integration efforts. Indeed, soccer has a long history of forging positive relationships across challenging divides, such as supporting social cohesion between Christians and Muslims in post-ISIS Iraq[19], social integration in postapartheid Africa[20] and peace-building in conflict zones such as Colombia and Northern Ireland[21].

Two forms of group alignment are likely to be particularly important in driving identity change among incarcerated people and encouraging society at large to support re-integration efforts. One is group identification[22,23], based on the sharing of particular beliefs, practices, clothes, hairstyles and other identity markers with members of a common category—such as sports clubs, religious organization, country or ethnic group[24,25]. Another form of group alignment is identity fusion[26,27], which is rooted in familial ties resulting from feelings of shared experience or common ancestry[28–30]. These two forms of group alignment entail contrasting relationships between personal and group identities[31–34]. In the case of identification, when the group identity is salient, the personal self becomes less accessible—leading to depersonalization[35] and de-individuation[36]. In the case of fusion, personal and group identities are instead activated synergistically, so that progroup action taps into personal agency and the personal self is felt to be strengthened by the power of the group[27,33]. Both identification and fusion prompt progroup behaviour but fusion is associated with much stronger forms of self-sacrifice[37] and lifelong loyalty[38].

If the aim is to foster long-term changes in identity among people in prison, by attaching them irrevocably to law-abiding groups and values—even when the temptations of recidivism are strong—then ideally interventions would try to bring out fusion with a positive community, such as that provided by the Twinning Project. Fusion has its strongest effects on close, relational ties[30,39], mirroring the strong bonds typical among family members[40] and so we would expect it to be most effective in bonding incarcerated people to each other or to the providers of Twinning Project courses—such as coaches. On the other hand, the Twinning Project is also a nationwide organization—a group category with which individuals may come to identify. Such forms of identification may provide a foundation on which fusion can build—combining both forms of group alignment to create what has been dubbed 'extended fusion'[27,41]. Extended fusion involves the activation of categorical ties to large-scale group identities but it does so on the basis of shared, personally self-defining experiences (for example, in nations at war[42,43]) or shared biological traits (for example, among members of an ethic group claiming shared ancestry[44]). As such, both fusion and identification have the potential to contribute to identity change for people in prison but also in positive attitude change in receiving communities.

Here we report the results of a unique quasi-experiment, contrasting behaviour between Twinning Project participants and a control group in the United Kingdom, in which we compare adjudications, case notes and self-harm incidents between the two samples using data shared by His Majesty's Prison Service (HMPS) (study 1). Adjudications refer to alleged disciplinary offences in prison which require an official hearing, typically capturing misconduct that is deemed dangerous to other inmates or staff or otherwise violates institutional policies[45]. The reporting of alleged offences can be contingent on staff-to-inmate ratios or prison policing styles[46], which are often viewed by incarcerated people as unfair and reflecting policing biases by prison staff[47]. Nevertheless, records of prison misconduct[48,49] and adjudications in UK prisons, have proved to be relatively reliable predictors of future re-offending[50] when triangulated with other evidence. We further examine the treatment group's social bonds and future optimism over a 5–8 month period (study 2). Optimism about desistance from crime is a well-established factor of re-offending in the criminological literature[51–53]. We also longitudinally analyse case notes; that is, qualitative observations by prison officers capturing positive or negative behaviours. Although we hope that increased optimism from participation in the Twinning project might positively impact re-offending rates, we are also aware that this cohort may not have appropriate strategies to act on their hope and thus be at a greater risk of disappointment, which could have knock-on effects for successful desistance from crime[52]. Triangulating our approach, we include the results of an online survey with formerly incarcerated people in the United Kingdom and the United States (study 3). Finally, without supportive receiving communities, the positive effects of prison-based interventions are unlikely to have lasting impacts following release. As such, we conducted a series of online studies with receiving communities in the United Kingdom and the United States to explore the relationships between social identities, perceived rehabilitation and community acceptance (studies 4–9).

## Results

### The impact of Twinning Project on prison behaviour

First, we analysed behavioural data from a cohort of people serving custodial sentences in 45 UK prisons who were enrolled on an intervention designed to reduce re-offending via soccer-based programmes with the prison's local major soccer club (studies 1 and 2). Participants attended 5–12 regular sessions with a coach from the club, often the biggest brand in the region, which led to an accredited qualification on completion of the programme. We compared indicators of prison behaviour (number of adjudications, positive and negative case notes and self-harm incidents) in a 2 month period after the programme between intervention participants ($n = 676$) and a control group ($n = 1,874$), which was matched for demographics (age and ethnicity), criminal history (index offence and Copas rate, that is, the rate of convictions over the length of a criminal career), incarceration details (prison category, prison behaviour assessment/incentives and earned privileges (IEP) level) and pretreatment prison behaviour (adjudications, case notes and self-harm).

We found that participation in the programme predicted significantly fewer proven adjudications (offences committed in prison), unstandardised beta (B) = −0.16, s.e. = 0.08, 95% confidence interval (CI) = −0.311 to −0.008, $F(1, 2,548) = 4.26$, $P = 0.039$, even when including matching parameters as covariates ($B = −0.15$, s.e. = 0.08, 95% CI = −0.303 to −0.007, $P = 0.040$; $F(28, 1,341) = 3.64$, $P < 0.001$). For every 100 incarcerated people, 15 Twinning Project participants had a proven adjudication in the 2 months following the course, whereas 31 control cases would receive an adjudication this period (Fig. 1). Other significant predictors of adjudications included more pretreatment adjudications ($B = 0.21$, s.e. = 0.08, 95% CI = 0.061–0.36, $P = 0.006$) and self-harm incidents ($B = 0.29$, s.e. = 0.13, 95% CI = 0.023–0.548, $P = 0.033$), more dense criminal histories (Copas, $B = 0.07$, s.e. = 0.03, 95% CI = 0.006–0.125, $P = 0.030$), as well as certain index offence categories (most strongly; violence against the person, $B = 0.30$, s.e. = 0.08,

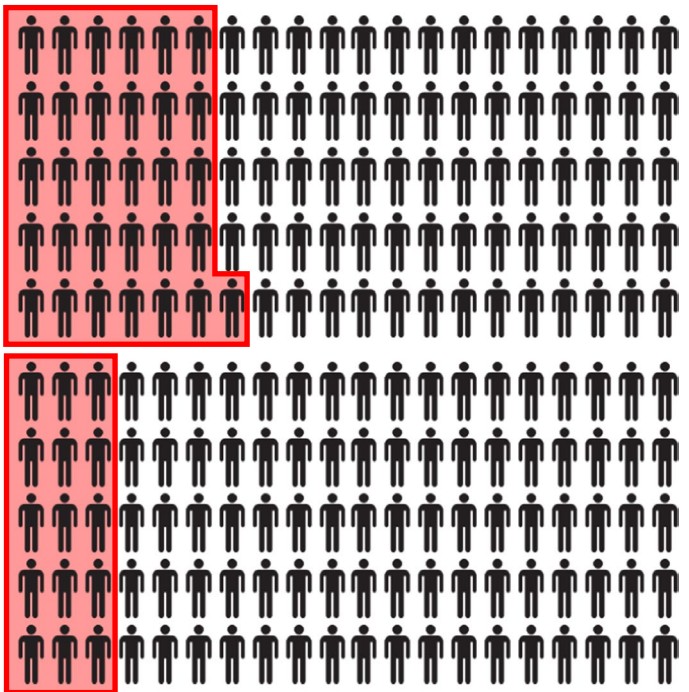

**Fig. 1 | Average number adjudications received per 100 people in prison in the 2 months after the programme.** Shown are the control group (top) and the treatment group (bottom). Visual representation of the control group (the intercept) and treatment group coefficient (intercept – estimated treatment effect, see intent-to-treat model 1 in Table A4 in Supplementary Information Section A).

95% CI = 0.133–0.460, $P < 0.001$; drug offences, $B = 0.29$, s.e. = 0.13, 95% CI = 0.079–0.299, $P = 0.001$). We found no significant treatment effects on other indicators of prison behaviour made available to us (case notes ($B = 0.17$, s.e. = 0.17, 95% CI = −0.167–0.506, $F(1,1,497) = 0.98$, $P = 0.323$) and self-harm incidents ($B = −0.02$, s.e. = 0.02, 95% CI = −0.053–0.009, $F(1,2,548) = 1.89$, $P = 0.170$)). We provide a detailed description of the matching procedure and treatment effect analyses, as well as extensive sensitivity analyses, throughout which the findings were robust, in Supplementary Information Section A.

### Social bonding and prison behaviour

To further understand how the programme worked, we examined the interplay of social-psychological and behavioural changes among intervention participants, drawing on longitudinal survey data from 19 prisons selected to be representative of regions in the United Kingdom and the categories of prisons involved in the programme ($n = 388$). The survey captured, among other things, participants' social bonding with different target groups and their optimism to succeed after release (see section on 'Materials' later and Supplementary Information Section B for measures). First, we conducted a series of paired samples $t$-tests to see if bonding to the Twinning Project developed over the course of the programme. Normality assumptions were violated but sensitivity analysis using the Wilcoxon signed-rank test showed that the results did not change substantively, ensuring robustness despite non-normal data. We report one-tailed $P$ values due to our directional hypotheses. Mean levels ($M$) of identification with the Twinning Project showed a small (Cohen's $d = 0.33$) but significant increase from the start (T0) ($M_{T0} = 3.63$) to the end (T1) of the programme ($M_{T1} = 4.00$), $t(243) = −5.08$, $P < 0.001$, and this increase remained significant at a follow-up 2 months later (T2) ($M_{T2} = 3.96$), $t(186) = −4.09$, $P < 0.001$, $d = 0.30$. Similarly, levels of identity fusion (with the Twinning Project) increased significantly from T0 ($M = 3.29$) to T1 (3.74), $t(255) = −4.96$,

$P < 0.001$, $d = 0.31$ but levels decreased slightly at the follow-up ($M_{T2} = 3.49$), such that the gain compared to the start of the programme was not statistically significant, $t(192) = −1.65$, $P = 0.050$, $d = 0.12$. We also tested if bonding to criminals would decrease over time and found no changes in levels of identification (T0–T1 $t(254) = −1.24$, $P = 0.108$, $d = 0.08$; T0–T2 $t(198) = −0.39$, $P = 0.349$, $d = 0.03$), while fusion to criminals slightly increased ($d = 0.13$) at the end of the programme, T0 ($M = 1.69$) to T1 (1.86), $t(260) = −2.12$, $P = 0.018$, but was no longer significantly higher at the follow-up survey, T0 ($M = 1.66$) to T3 (1.77), $t(199) = −1.24$, $P = 0.108$, $d = 0.09$.

Next, to analyse within-individual changes more thoroughly, we looked at prison behaviour before (pre) and after (post) the intervention amongst those who undertook the programme. Normality assumptions for paired samples $t$-tests were violated but results of Wilcoxon signed-rank test confirmed robustness of results. Average levels remained stable at desirable levels for the entire sample, that is, low levels of adjudications, self-harm incidents and a positive case note balance (adjudications, $M_{pre} = 0.13$–$M_{post} = 0.15$, $t(833) = −1.28$, $P = 0.101$, $d = 0.04$; case notes, $M_{pre} = 0.66$–$M_{post} = 0.48$, $t(675) = 1.57$, $P = 0.058$, $d = 0.06$; self-harm, $M_{pre} = 0.01$–$M_{post} = 0.02$, $t(833) = 0.58$, $P = 0.249$, $d = 0.02$), reflecting the fact that Twinning Project participants are often particularly 'well-behaved'. However, among those with at least one adjudication before the intervention, significant improvements were observed, $M_{pre} = 1.30$–$M_{post} = 0.48$, $t(80) = 7.23$, $P < 0.001$, $d = 0.80$. Using logistic regression analysis, we further explored whether improved behaviour (decreased number of adjudications after treatment) was associated with positive changes to identification with the Twinning Project and found a significant effect of identification change on behaviour improvement ($B = 0.63$, s.e. = 0.29, OR = 1.88, 95% CI = 1.058–3.345, $P = 0.031$; $\chi^2(1) = 4.61$, $P = 0.032$ Nagelkerke $R^2 = 0.08$). The effect remained significant ($B = 2,15$, s.e. = 1.03, OR = 8.58, 95% CI = 1.139–64.624, $P = 0.037$; $\chi^2(11) = 51.09$, $P < 0.001$ Nagelkerke $R^2 = 0.76$). even after controlling for baseline prison behaviours, age, prison type, Copas rate and time until release. We report full model results in Supplementary Information Section B.

Examining participants' future optimism about their employability and chances to desist, we found that, similar to prison behaviour, optimism across the entire sample was either stable or showed small improvements at very desirable (high) levels (optimism about employability, $M_{T0} = 4.09$–$M_{T1} = 4.14$, $t(266) = −1.10$, $P = 0.137$, $d = 0.07$ and $M_{T0} = 4.06$–$M_{T2} = 4.23$, $t(204) = −3.01$, $P = 0.001$, $d = 0.21$; optimism about desistance, $M_{T0} = 4.26$–$M_{T1} = 4.27$, $t(253) = −0.15$, $P = 0.439$, $d = 0.01$ and $M_{T0} = 4.36$–$M_{T2} = 4.35$, $t(196) = 0.17$, $P = 0.431$, $d = 0.01$). Among participants whose baseline levels were not already at the ceiling level, we observed significant boosts to optimism about both outcomes by the end of the programme and until the follow-up survey (optimism about employability, $M_{T0} = 3.61$–$M_{T1} = 3.91$, $t(174) = −5.06$, $P < 0.001$, $d = 0.38$ and $M_{T0} = 3.62$–$M_{T2} = 4.06$, $t(138) = −6.62$, $P < 0.001$, $d = 0.56$; optimism about desistance, $M_{T0} = 3.66$–$M_{T1} = 4.01$, $t(139) = −5.46$, $P < 0.001$, $d = 0.46$ and $M_{T0} = 3.75$–$M_{T2} = 4.10$, $t(101) = −4.26$, $P < 0.001$, $d = 0.42$). Increased optimism about one's capacity to find employment ($r = 0.16$, 95% CI = 0.031–0.280, $P = 0.015$) and stay out of trouble ($r = 0.24$, 95% CI = 0.111–0.358, $P < 0.001$) also correlated with increased identification to the Twinning Project. We report full correlation tables in Supplementary Information Section B.

### Bonding and behaviour among formerly incarcerated people

To see whether these findings translate to 'the outside' and how social bonds to a variety of targets relate to desistance, we captured data from formerly incarcerated people in an online study in the United Kingdom and the United States (study 3, $n = 250$). We conducted multiple linear regression analyses, estimating robust standard errors to correct for heteroscedasticity. The results suggested that social bonds with a criminal group are both directly and indirectly linked to procriminal attitudes. Fusion to criminals or friends involved in

crime significantly and positively correlated with stronger procriminal attitudes ($B = 1.23$, s.e. $= 0.27$, 95% CI $= 0.694–1.767$, $P < 0.001$), whereas people fused to their country held weaker criminal attitudes ($B = −0.93$, s.e. $= 0.30$, 95% CI $= −1.526$ to $−0.337$, $P = 0.002$, $F(10,194) = 6.39$, $P < 0.001$, $R^2 = 0.25$). Fusion to criminals also predicted more frequent instances of self-reported criminal behaviour ($B = 0.27$, s.e. $= 0.06$, 95% CI $= 0.102–0.322$, $P < 0.001$, $F(10,131) = 5.06$, $P < 0.001$, $R^2 = 0.28$), arrest ($B = 0.24$, s.e. $= 0.07$, 95% CI $= 0.092–0.388$, $P = 0.002$, $F(10,155) = 3.75$, $P < 0.001$, $R^2 = 0.20$) and reconviction after release from prison ($B = 0.22$, s.e. $= 0.07$, 95% CI $= 0.091–0.351$, $P = 0.001$) (all models control for age, gender, nationality, relationship status, number of children, employment, socio-economic status and social desirability). We report full model results in Supplementary Information Section C. We further tested whether the effects of identity fusion were dependent on (moderated by) individuals' beliefs about a group's values (for example, being supportive, law-abiding and honest) and whether future optimism could help explain (mediate) the relationships with criminal behaviours or attitudes. We found that the effect of fusion to one's country was conditional on whether the individual associated their country with support, such that individuals who did not perceive their country as supportive held stronger procriminal attitudes (fusion×value support, $B = 1.10$, s.e. $= 0.45$, 95% CI $= 0.212–1.998$, $P = 0.016$; $F(24,176) = 4.53$, $P < 0.001$, $R^2 = 0.38$). Furthermore, using Hayes' (2022)[54] PROCESS macro (model 4) we found a significant indirect effect ($B = 0.14$, s.e. $= 0.08$, 95% CI $= 0.018–0.334$) of fusion to criminals on procriminal attitudes via the belief that one could stay out of trouble with the law, $F(11,193) = 6.12$, $P < 0.001$, $R^2 = 0.26$ (Fig. 2). We report full model results in Supplementary Information Section C.

## Re-integration support among receiving communities

Having established that social ties relate to criminal behaviour and attitudes, both inside (studies 1 and 2) and outside prison (study 3), we focussed on how social bonding mechanisms might relate to receiving attitudes of communities towards formerly incarcerated people. Specifically, considering the challenges in postprison employment, we conducted online studies in the United Kingdom and the United States which investigated willingness to hire a formerly incarcerated person for a job, as well as perceived chances of getting a good job and staying out of trouble with the law in multiple contexts (Table 1). Participants, all of whom had hiring experience, were invited to consider a job application vignette whereby they evaluated applicants with criminal backgrounds. In study 4, we recruited a sample of British soccer fans and non-fans ($n = 234$) to understand how interventions might influence public perceptions and willingness to hire formerly incarcerated people. Participants were presented with three applicants who had completed a soccer programme (the Twinning Project), a gardening programme or completed no programme in prison and they rated how likely they would be to hire each candidate (employability) and how good their chances were to stay out of trouble with the law (future chances). Repeated measures analyses of variance (ANOVAs) were conducted to compare mean levels (M) between groups and, because of violation of the sphericity assumption, Greenhouse–Geisser corrected results are reported. Compared to applicants who did not complete a programme ($M_{employability} = 2.68$, s.e. $= 0.07$; $M_{future chances} = 1.60$, s.e. $= 0.03$), formerly incarcerated people who had completed an educational programme, such as the Twinning Project ($M_{employability} = 4.38$, s.e. $= 0.06$; $M_{future chances} = 2.91$, s.e. $= 0.03$) or a vocational gardening programme ($M_{employability} = 4.38$, s.e. $= 0.06$; $M_{future chances} = 2.72$, s.e. $= 0.03$), were rated as significantly more employable, $F(1.63, 377.74) = 373.20$, $P < 0.001$, $\eta^2 = 0.617$, and more likely to desist from crime, $F(1.66, 385.14) = 865.08$, $P < 0.001$, $\eta^2 = 0.789$; $F(1.66, 385.14) = 865.08$, $P < 0.001$, $\eta^2 = 0.789$. No evidence supported the between-group effect of fan status ($F_{employability}(1,232) = 1.08$, $P = 0.301$; $F_{future chances}(1,232) = 2.33$, $P = 0.128$). Likewise, the interactions between fan status and programme type were non-significant ($F_{employability}(1.63, 477.74) = 0.78$, $P = 0.435$;

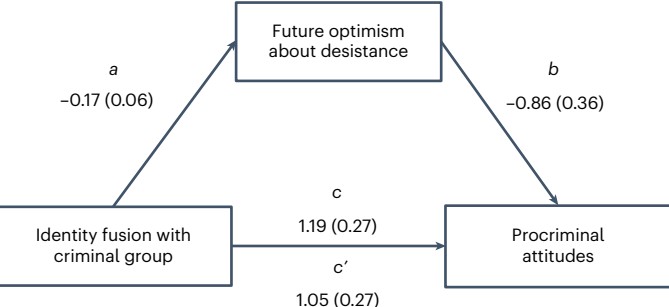

**Fig. 2 | Social bonding with criminals both directly and indirectly (via perceived chances to desist) contributes to procriminal attitudes.** In the mediation diagram $a$, $b$, $c$ and $c'$ are path coefficients representing unstandardized regression weights and robust standard errors (in parentheses). The $c$ path coefficient represents the total effect of fusion to a criminal group on procriminal attitudes. The $c'$ path coefficient refers to the direct effect of criminal group fusion on procriminal attitudes. [95% CI] and $P$ values: $a = [−0.294, −0.039]$, $P = 0.011$; $b = [−1.574, −0.152]$, $P = 0.018$; $c = [0.667, 1.721]$, $P < 0.001$; and $c' = [0.509, 1.592]$, $P < 0.001$ (Table C4 in Supplementary Information Section C).

$F_{future chances}(1.66, 385.14) = 1.90$, $P = 0.158$, suggesting that sports identity was not a powerful motivator of re-integration acceptance in its own right and that participants valued the two educational programmes similarly. Correlational analyses further showed that willingness to hire ($r = 0.60$, 95% CI $= 0.510–0.675$, $P < 0.001$), as well as perceived future chances ($r = 0.65$, 95% CI $= 0.564–0.725$, $P < 0.001$), positively correlated with a perception that an applicant had gained transferrable skills through their programme (full model results are reported in Supplementary Information Section D).

To achieve high ecological validity, we elaborated on the vignette design and invited participants to rate applicants with criminal histories again, this time under the premise of conducting a social media background check as is commonly performed in real-life hiring scenarios (studies 5–7). Participants were asked to form an impression based on a candidate's social media posts and again rated the applicant's employability and chances of staying out of trouble with the law. We measured fusion to the applicant and other known correlates of lay attitudes towards formerly incarcerated people, including political ideology, open-mindedness, interpersonal contact with formerly incarcerated people and demographic factors (socio-economic status, age and education). Multiple linear regression analysis showed that for British soccer fans (study 5, $n = 303$) fusion with the applicant was the strongest predictor of willingness to hire ($B = 0.38$, s.e. $= 0.06$, 95% CI $= 0.251–0.502$, $P < 0.001$; $F(11, 290) = 7.48$, $P < 0.001$, $R^2 = 0.22$) and perceived chances to desist from crime ($B = 0.24$, s.e. $= 0.05$, 95% CI $= 0.149–0.335$, $P < 0.001$; $F(11, 290) = 4.12$, $P < 0.001$, $R^2 = 0.14$), beyond other well-researched criminal justice predictors. The results of study 5 replicated with a sample of US American football fans (study 6, $n = 294$) where fusion predicted willingness to hire ($B = 0.35$, s.e. $= 0.06$, 95% CI $= 0.227–0.464$, $P < 0.001$; $F(11, 281) = 10.08$, $P < 0.001$, $R^2 = 0.28$) and perceived to desist from crime ($B = 0.35$, s.e. $= 0.05$, 95% CI $= 0.243–0.454$, $P < 0.001$; $F(11, 281) = 6.42$, $P < 0.001$, $R^2 = 0.20$) above all other covariates and with a sample of US citizens (study 7, $n = 319$) where identity fusion outperformed all other predictors of willingness to hire ($B = 0.31$, s.e. $= 0.05$, 95% CI $= 0.193–0.423$, $P < 0.001$; $F(11, 304) = 8.13$, $P < 0.001$, $R^2 = 0.22$) and perceived chances of desistance ($B = 0.26$, s.e. $= 0.05$, 95% CI $= 0.165–0.350$, $P < 0.001$; $F(11, 304) = 5.75$, $P < 0.001$, $R^2 = 0.17$). Full model results are reported in Supplementary Information Sections E–G. Finally, we replicated these findings in both the United Kingdom and the United States using an alternative methodology to check that any biases towards other social media users were not unduly influencing the results. Here, participants were asked to rate applicants on the basis of information gathered in a

**Table 1 | Breakdown of demographics for studies 3–9**

| Study—sample | n | Age (years) | | Gender | | | | Ethnicity | | | | | | |
|---|---|---|---|---|---|---|---|---|---|---|---|---|---|---|
| | | M | s.d. | Male | Female | Other | Unknown | White | Black | Asian | Mixed | Hispanic | Other | Unknown |
| 4—UK nationals | 234 (46.6% soccer fans) | 44.52 | 14.05 | 65.4% | 34.2% | 0.4% | 0% | 90.2% | 2.1% | 5.1% | 2.1% | n/a | 0% | 0.4% |
| 5—UK soccer fans | 303 | 41.12 | 14.25 | 80.5% | 18.8% | 0% | 0.7% | 82.5% | 1.7% | 9.2% | 4% | n/a | 1.3% | 1.3% |
| 6—US American football fans | 294 | 40.92 | 12.47 | 96.9% | 2.4% | 0.3% | 0.3% | 76.2% | 7.1% | 3.7% | 4.1% | 7.1% | 1.4% | 0.3% |
| 7—US nationals | 319 | 42.49 | 11.69 | 41.4% | 57.7% | 0.6% | 0.3% | 74.0% | 7.8% | 5.0% | 3.4% | 6.3% | 2.5% | 0.9% |
| 8—UK nationals | 327 | 42.09 | 11.17 | 54.7% | 44.6% | 0% | 0.6% | 92.0% | 0.6% | 4.9% | 1.8% | n/a | 0% | 0.6% |
| 9—US nationals | 320 | 44.84 | 13.06 | 56.6% | 42.5% | 0.6% | 0.3% | 82.5% | 5.0% | 6.3% | 1.6% | 3.1% | 1.3% | 0.3% |

Note, n/a indicates that data were not collected for that specific study.

job interview where the applicant explained their motivation to stay out of trouble with the law. Again, fusion to the applicant was the key predictor of willingness to hire in both the United Kingdom (study 8, $n = 335$, $B = 0.31$, s.e. $= 0.05$, 95% CI $= 0.221–0.398$, $P < 0.001$; $F(10, 315) = 11.35$, $P < 0.001$, $R^2 = 0.27$) and the United States (study 9, $n = 332$, $B = 0.44$, s.e. $= 0.05$, 95% CI $= 0.351–0.528$, $P < 0.001$; $F(10, 308) = 18.04$, $P < 0.001$, $R^2 = 0.37$), beyond other relevant criminal justice indicators (full model results are reported in Supplementary Information Sections H and I).

## Discussion

We found that the Twinning Project, a soccer-based prison intervention with prominent social identities attached to it, had a positive impact on adjudications, compared to a control group. Adjudications, hearings for offences committed within a prison, are not uncommon in UK prisons and equivalents may be found in most prison systems around the world; they are arguably the most objective measure of prison behaviour available and, coupled with self-reported future optimism, provide a basis for forecasting re-offending rates associated with interventions[50]. Across nine studies, social bonding was the key factor in predicting a series of variables which the criminal justice system relies upon for reduced crime rates: improved prison behaviour, sustained desistance from crime when leaving prison and positive attitudes towards formerly incarcerated people in the receiving community, including willingness to hire for a job. The nuances of social bonding offer a helpful framework for understanding the success of high-profile interventions such as the Twinning Project.

Results indicated that social identification, rather than identity fusion, was associated with decreased adjudications in prison. Relatedly, identification increased during the intervention and fusion remained relatively stable. Social identification is clearly powerful and plays a role in creating cohesive social structures among large groups of depersonalized individuals[22,35]. We propose that the intervention is a critical first step in creating longer-term changes; by investing in more indepth programmes that have space for participants to integrate emergent fusion with one another and the project, there may be opportunities to improve associated behaviours further. For instance, we found that fusion did increase following the programme but that this change had reverted towards baseline 2 months after the programme.

Fusion to criminal groups was a significant correlate of procriminal attitudes among formerly incarcerated people. Here, fusion to criminal groups was associated with less confidence to stay out of trouble with the law, while associations with extended positive group identities, such as one's nation, were linked to law-abiding attitudes. This suggests that extended positive group identities such as the Twinning Project or other interventions, if invested in while in prison, could become pivotal group targets that not only address prison behaviour but a sustained commitment to legal activity. However, of several seemingly positive targets, only fusion to nation predicted better behaviours, presumably because some social groups that may be a positive influence on many of us also have the potential to be a negative influence, as exemplified by major crime families[52]. Interestingly, fusion to more tangible social groups (for example, family and friends) was weakly or ambiguously associated with desistance outcomes, suggesting that smaller (relational) and larger (extended) groups play unique roles in the re-integration experience and outcomes of formerly incarcerated people.

Coupled with the effects of social cohesion, our results suggest that engagement in activities that reflect group/societal norms (for example, completing education) elicit re-integration support among the receiving community. Perhaps surprisingly, soccer fan identities did not instil particularly positive views towards soccer programme alumni among the receiving community. Despite previous research into the import of soccer identities in an array of group-oriented behaviours[31,38,55,56], soccer identities may not be sufficient to evince trust and inclusion over larger superordinate identities, such as being an educated worker, which may have been primed through the experiment. Instead, more open-mindedness, a belief that people can change their ways, was associated with more optimism about re-integration among the receiving community. However, even for particularly open-minded participants, this only held when the formerly incarcerated person had a credible history of transformative behaviour; that is, they had completed an educational programme in prison. Taken together, these results offer a very human perspective, highlighting the value of social identity, relational ties and feelings of connection in overcoming stigma.

In public discourse on the global re-offending crisis, attention is often focused on the Global North—specifically countries with more developed economies and, in the case of our study, highly individualistic norms (the United Kingdom and the United States). We predict that attitudes among receiving communities will be highly variable and encourage researchers to investigate these cultural nuances. For instance, in Israel, a relatively sympathetic stance towards formerly incarcerated people has been found[57], which may be explained by ethnic and religious identities effectively trumping lower-order social categories such as 'criminal' in a nation where higher-order identities are particularly prominent as a result of wider socio-political circumstances.

We also note that our prison study was conducted shortly after Covid-19 ceased to be a national emergency in the United Kingdom, which may have biased results in several ways. Importantly, recruitment

for the Twinning Project favoured relatively well-behaved and optimistic participants, such that opportunities for behavioural and attitudinal improvements were relatively limited in our sample. Consequently, significant reductions in adjudications and increases in optimism were observed only among participants who previously got into trouble and who did not enter the programme with excellent confidence in their future. With more diverse and inclusive cohorts that reflect the true composition of prison populations, there could be an opportunity for greater improvements in observed outcomes. We encourage future research to further investigate what we see as the great potential of shared, transformative experiences to bond people to the collective and the related effects on socially desirable behaviours.

Our findings have implications for prison policy, practice and future research. The positive impact of the Twinning Project on adjudications compared to a control group underscores the significance of social bonding in shaping behaviours and attitudes among incarcerated people. This emphasizes the need to invest in interventions that foster group identities among incarcerated individuals, as good prison behaviour makes the prison estate a more positive place to live and work for both inmates and staff[1]. Others paved the way for academic research into prison service sports programmes and the potential to turn the lives of people in prison around. For instance, sports interventions provide an opportunity for physical activity, with associated positive effects on dopamine, mood regulation, sustained concentration and a host of physical and mental wellbeing factors likely to play into desistance behaviours[58,59]. Additionally, sports can provide a potentially unparalleled locus for much-needed social connections, as our evidence on the effects of the Twinning Project clearly demonstrates.

Additionally, with a proven effect on adjudications, our data also suggest that there may be potential for economic savings associated with such programmes (that is, less solitary confinement and fewer adjudication hearings which may in turn contribute to shorter sentences). We also predict that the present findings will relate to reduced re-offending rates for Twinning Project participants, data that will be analysed after the current cohort has been released. However, while social identification is a critical first step, the study supports further investment in programmes allowing participants to reap the benefits of emergent identity fusion for positive long-term behavioural outcomes. Moreover, the study highlights the complex role of group identities, indicating that, although fusion to positive group identities such as the nation correlates with better behaviours, fusion to criminal groups may undermine law-abiding attitudes. This reflects the need for policy to support interventions that promote positive societal norms and values, facilitating successful re-integration. Precisely who determines these norms is a matter of debate and needs careful planning, ideally consulting with those directly affected by crime, including victims, perpetrators and their families[60]. More research is needed to understand the cultural nuances that shape attitudes towards formerly incarcerated individuals both within and beyond the contexts studied here.

The allure of major sports clubs and brands to solve global crises lies not only in the billions of dollars in revenue they may contribute to social issues but in their billions of loyal fans. Our data suggest that soccer fandom offers a powerful pathway to more prosocial, law-abiding identities for people in prison but that a broad range of educational interventions are needed to help tackle prison stigma among receiving communities. Our findings suggest that interventions combining a strong social element and educational appeal for the public are urgently needed to help address the global prison crisis.

## Method

### Ethics and preregistration
Studies 1 and 2 were approved by the ethics board of the University of Oxford (SAME_C1A_19_016) and the National Research Committee (2019-215). Studies 3 and 8 (ethics ID: 20231674482788242),

studies 4 and 5 (ethics ID: 20231675692193254), study 6 (ethics ID: 20231679036795309), study 7 (ethics ID: 20231680606154343) and study 9 (ethics ID: 20231678778665296) were approved by the ethics board of the School of Anthropology and Conservation at the University of Kent. Informed consent was obtained from all participants who took part in survey research (studies 2–9) and the processing of incarcerated people's personal data without explicit consent was in accordance with the Data Protection Act 2018 (Schedule 1, ¶6–28).

Studies 1 and 2 were preregistered via OSF as part of a long-term project evaluating the impact of Twinning Project on re-offending (https://osf.io/2f4zg, 20 April 2023). Studies 3–9 were preregistered (or conceptual replications of preregistered studies) as part of a larger project examining social bonding among formerly incarcerated people and receiving communities (23 February to 16 May 2023), study 3 (https://osf.io/gmxuj), study 4 (https://osf.io/t6r3q), studies 5–7 (https://osf.io/nt2e4) and studies 8 and 9 (https://osf.io/x8dz2). Deviations from preregistered analyses plans are disclosed in the Results and originally planned analyses are reported in the Supplementary Information.

### Data collection
Data for studies 1 and 2 were collected from the population of HMPS UK (men and women), including all people enroled in the Twinning Project (treatment group) between September 2021 and March 2023 and a matched control group. For these analyses, we worked exclusively with the male population because of the unique needs of the women's population and highly unbalanced sample sizes (women make up around 5% of the prison population). For study 1, $n = 1,411$ individuals were initially identified as potential Twinning Project participants within the research period and the data of $n = 834$ eligible male participants were obtained from HMPS. Cases from private institutions ($n = 158$) were excluded from the main analyses because of inconsistent data recording procedures (Supplementary Information Section A) and the final treatment group sample consisted of $n = 676$ people in prison (Mean ($M_{age}$) = 31.08, s.d.$_{age}$ = 7.49, White = 58%, Black = 22%, Asian = 8%, mixed = 10%, other = 1%, unknown = 1%) and the control group sample consisted of $n = 1,874$ people in prison (weighted averages; $M_{age}$ = 31.04, s.d.$_{age}$ = 7.74, White = 63%, Black = 16%, Asian = 8%, mixed = 12%, other = 1%). Of the treatment group, $n = 55$ did not complete the course (9.2% attrition) and separate analyses with this reduced sample are provided in Supplementary Information Section A. The average initial cohort size was 13.06 participants (s.d. = 3.80, range 6–24) and the average programme length was 6.07 weeks (s.d. = 3.31, range = 1–19). For study 2, a subsample of $n = 388$ male participants completed the first wave of the longitudinal survey at T0 ($M_{age}$ = 30.20, s.d.$_{age}$ = 7.12, White = 60%, Black = 20%, Asian = 9%, mixed = 10%, other = 2%) (T1, $n = 283$; T2, $n = 213$).

No statistical methods were used to predetermine sample sizes but our sample sizes are larger than those reported in previous publications with comparable study design (for example, refs. 61,62) and larger than samples required for propensity score model approaches to replicate randomized control trial results[63]. There were no randomized elements in the assignment to treatment groups or data collection for studies 1 and 2 which reflect restrictions in the programme delivery and data collection. Further details on the recruitment methods and data processing are presented in Supplementary Information Sections A and B.

Data for studies 3 to 9 were collected using the online crowdsourcing platform Prolific. Individual samples from the United Kingdom and the United States were recruited between January and May 2023 and participants received on average the equivalent of £9.00 per hour in financial reimbursement for taking part. Sample sizes were determined by a priori power analyses in G* Power[64] to detect small–moderate effect sizes in linear regression/ANOVA analyses ($f^2$ = 0.08/0.09, $f$ = 0.18) at the standard 0.05 error probability at 0.80 power. Demographics can be found in Table 1. Details on data processing and sample demographics are presented in Supplementary Information Sections C–I.

Data of individual participants were removed on the basis of data quality checks (for example, missing attention checks). The number of exclusions was: study 3, $n = 1$; study 4, $n = 48$; study 5, $n = 28$; study 6, $n = 36$; study 7, $n = 14$; study 8, $n = 8$; and study 9, $n = 10$. Data collection and analysis were not performed blind to the conditions in studies 1 and 4.

## Materials

Study 1 materials reflect the data typically captured within British prisons, shared with us by HMPS. The key outcome variables were observational recordings of adjudications, self-harm incidents and positive/negative case notes, as well as demographics (age and ethnicity) and criminal history variables[65] (index offence, prison behaviour assessment and IEP levels).

Study 2 materials reflect part of an original longitudinal pen-and-paper survey distributed among a subsample of Twinning Project participants at the beginning, end and 2 months after the programme, distributed by prison and soccer club staff involved in the programme. Survey items were answered using a 1 = strongly disagree to 5 = strongly agree Likert-scale format, unless specified otherwise. Survey measures captured, among other things, participants' social bonding (social identification[66], 'I identify with [x]'; pictorial identity fusion[26], 'Overlap between circles of self and group', A = no relationship to E = total oneness) with reference to different groups (Twinning Project, family, soccer fans, country and a criminal group) and future optimism about employment and desistance from crime ('What do you think your chances are to have/keep a good job?/stay out of trouble with the law?', 1 = poor to 5 = excellent).

Studies 3–9 were online surveys conducted with Qualtrics. Study 3 contained measures on future optimism, social bonding (pictorial identity fusion) to different groups (for example, family, close friends, a criminal group and my country), as well as time spent (days per week) and values associated with the respective groups (honest, law-abiding and supportive, 1 = not at all to 4 = a lot). The survey also measured self-reported criminal behaviour since release from prison (ten items, for example, traffic offence, theft and assault, 1 = never to 5 = often, Cronbach's $\alpha = 0.84$) and procriminal attitudes[67] (nine items, for example, 'Successful people break the law to get ahead', 1 = disagree to 3 = agree, Cronbach's $\alpha = 0.80$). The study contained additional control variables, including demographics (age, ethnicity, education, nationality, relationship status, parenthood status and employment status), social desirability[68] (16 items, for example, 'I always admit my mistakes open and face the potential negative consequences', 1 = true to 0 = false, mean Cronbach's $\alpha = 0.76$) and subjective socio-economic status[69] ('Where would you put yourself on the ladder?', 1 = worst off to 10 = best off).

In study 4, each participant (in the role of a hiring manager) was presented with three descriptions (in random order) of male job applicants who served comparable sentences in prison and who successfully completed the Twinning Project, a gardening programme or no educational programme (reflecting the levels of an experimental condition). The two educational programmes were described with regard to their third-party certification, duration and basic learning outcomes. Participants completed measures of perceived transferrable skills of the applicant (eight items, 'To what extent does the applicant possess the following skill? for example, problem-solving', 1 = not at all to 4 = a lot, mean Cronbach's $\alpha = 87.7$), soccer fandom ('Do you consider yourself a football fan', 1 = yes, 0 = no), willingness to hire (1 = definitely unwilling to 6 = definitely willing) and the perceived future chances of the applicant ('How would you rate [x]'s chances to have/keep a good job?/ stay out of trouble with the law?', 1 = poor to 5 = excellent). The study also included a combination of control variables relevant for attitudes towards offenders (open-mindedness[70], eight items, for example, 'The kind of person someone is, is something very basic about them and it can't be changed very much', 1 = strongly disagree

to 6 = strongly agree, mean Cronbach's $\alpha = 0.96$), political ideology[71] (three items, for example, 'How would you place your views on this scale when you think about social issues?', 0 = left to 10 = right, mean Cronbach's $\alpha = 0.95$), contact with formerly incarcerated people[72] ('How many people do you know personally or professionally who have been to prison?', 1 = none to 4 = many), criminal victim status ('Have you or a family member ever been a victim of crime?', 1 = yes, 0 = no), hiring experience ('How many hiring decisions do you estimate you have been involved in?', 1 = none to 7 = more than 50), subjective socio-economic status and level of education.

Studies 5–9 measured participants' social bonding with a job applicant (verbal identity fusion, Gómez et al., 2011[73], four items, for example, 'I have a deep emotional bond with the applicant', 1 = strongly disagree to 7 = strongly agree, mean Cronbach's $\alpha = 87.2$), their willingness to hire, perceived future chances of the applicant (to desist from crime/to have a good job) and included the same control variables described for study 4 (studies 8 and 9 did not measure social desirability and perceived future chances). We list all measures and respective items (including internal reliability scores for multiitem measures), as well as additional contextual information provided to participants for all studies in Supplementary Information Sections A–I.

## Design and analyses

Study 1 used a quasi-experimental design to estimate the effects of being assigned to the Twinning Project compared to a matched control group, which is the standard approach in criminology research when a randomized control trial is not feasible. For study 1, we used the toolkit for weighting and analysis of non-equivalent groups (TWANG)[74] to balance treatment and comparison group responses by applying weights on the basis of gradient boosted regression models to estimate the average treatment effect of the treated[75]. Propensity score models were fitted to predict membership in the treatment group (Twinning Project) based on demographic factors (age and ethnicity), institutional factors (prison security level), as well as indicators of prison behaviour (pretreatment IEP level, adjudications, case note balance and self-harm incidents) and criminal history (index offence and Copas score), using a stopping method rule to minimize the mean standardized difference between groups. Copas score was included as a measure of offending density and is commonly used in the offender group reconviction scale, to control for group differences in re-offending likelihood[65,76]. Group balance was subsequently evaluated on the basis of statistical significance and size of standardized mean differences and distribution difference statistics. Treatment effects were estimated using generalized linear models predicting post-treatment prison behaviour on the basis of treatment group membership and additionally based on matching criteria to estimate doubly robust estimates. All steps were conducted twice, first following an intent-to-treat approach, which analyses the data of all cases admitted to the programme regardless of completion status, and following a protocol adherence approach, only including cases who completed the treatment. The preregistration did not specify a method for processing case-control data, expecting a 1:1 matched control sample, which was not provided. Instead, a larger, demographically similar control population was supplied. We chose the TWANG approach[74] for analysis, abandoning the preregistered independent sample *t*-tests. Preregistered analysis of 'activity/work attendance' and 'visit attendance' rates as outcome measures were also abandoned because of data quality issues. We report additional details on initial data processing, justifications for the chosen model parameters, as well as extensive sensitivity analyses in Supplementary Information Section A.

Study 2 was a correlational study drawing on longitudinal data. For this, we first conducted a series of paired sample *t*-tests to compare mean levels of prison behaviours and social bonding indicators over time. In line with preregistered hypotheses, we explored pre–post differences among treatment group members with baseline scores above floor and below ceiling levels. Subsequently, we computed difference

scores for variables where significant change was observed and we used both bivariate correlations and logistic regression analyses to estimate the relationship between social bonding change and prison behaviour change, while controlling for relevant covariates. Preregistered analyses to examine 'defusion' from criminal groups was deemed flawed and not conducted as the hypothesized decrease in fusion to criminal groups was not observed in the sample and analyses to predict postintervention levels of prison behaviour based on fusion to the Twinning Project showed no significant effects and are reported as part of Supplementary Information Section A.

Studies 3 and 5–9 were correlational, drawing on cross-sectional data. For these studies, we first estimated bivariate correlation coefficients to establish significant relationships between outcome and control variables and then conducted multiple linear regression analyses to estimate the effects of social bonding on indicators of re-integration support while controlling for relevant covariates. For study 3, we also used the PROCESS macro in SPSS, which uses multiple regression models to estimate mediation (model 4) and moderation (model 1) effects. The associations between identity fusion (to a job applicant) and willingness to hire or perceived future chances of the applicant in studies 5–9 were originally preregistered to be tested as part of larger models; that is, as a mediator between experimental triggers of fusion and re-integration support for the applicant. The evaluation of the experimental effects on identity fusion and indirect effects on re-integration support for a formerly incarcerated person are reported as part of another manuscript. The results reported here do not change when tested as part of the original model. Any overlap in reporting of results has been declared.

Study 4 was a 2 (fan status: soccer fan versus non-fan) × 3 (programme type: Twinning Project versus gardening versus no programme) mixed model experiment. Stimuli of the within-group factor (programme type) were presented in randomized order using block randomization. Repeated measure ANOVAs were conducted to estimate effects of programme type and fan status on evaluations of job applicants' employability, chances to succeed in life and skilfulness. A preregistered follow-up analysis to test whether skilfulness could explain differences in evaluations between fans versus non-fans was not conducted as there were no main effects of fan status.

Assumptions of the statistical tests were met and $P$ tests are two-tailed unless specified otherwise.

### Reporting summary
Further information on research design is available in the Nature Portfolio Reporting Summary linked to this article.

## Data availability
For S1-S2, prison data is controlled by HMPPS and requests to obtain it should be directed to HMPPS following National Research Committee (NRC) approvals, https://apply-for-hmpps-research.service.justice.gov. uk/Introduction-and-Guidance/. Survey data for this population, which was collected by the researchers, can be obtained from the research team on request.

Data that is not on OSF will be held by the researchers for 10 years as per standard University retention guidelines.

Datasets and output generated for studies 3–9 can be found on OSF at https://osf.io/u74bf/?view_only=d27b1d321455470d8dab9a 5ae0932029.

## Code availability
Codes for studies 3–9 can be found on OSF at https://osf.io/u74bf/?view_ only=d27b1d321455470d8dab9a5ae0932029. Code can be accessed via email requests to the lead author.

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

## Acknowledgements
This work was supported by an UKRI Future Leaders Grant (MR/T041099/1 to M.N.) and an advanced grant ('Ritual modes: divergent modes of ritual, social cohesion, prosociality and conflict', grant agreement no. 694986) from the European Research Council under the European Union's Horizon 2020 Research and Innovation Programme (to H.W.). The funders had no role in study design, data collection and analysis, decision to publish or preparation of the manuscript. We thank the Twinning Project and HMPS for their ongoing support of the research.

## Author contributions
M.N. and H.W. conceived the project and designed studies 1 and 2. M.N. and L.P. designed studies 3–9. L.P. and J.C. analysed the data. L.P. designed the figures and prepared the Supplementary Information. M.N. led the write-up and all authors contributed to writing.

## Competing interests
L.P. is a part-time research consultant for the Twinning Project. His role started after the submission of this manuscript and is unrelated to the data or analysis presented in this article. The other authors declare no competing interests.

## Additional information

**Correspondence and requests for materials** should be addressed to Martha Newson.

# Reporting Summary

## Statistics

For all statistical analyses, confirm that the following items are present in the figure legend, table legend, main text, or Methods section.

| n/a | Confirmed | |
|---|---|---|
| ☐ | ☒ | The exact sample size (*n*) for each experimental group/condition, given as a discrete number and unit of measurement |
| ☐ | ☒ | A statement on whether measurements were taken from distinct samples or whether the same sample was measured repeatedly |
| ☐ | ☒ | The statistical test(s) used AND whether they are one- or two-sided<br>*Only common tests should be described solely by name; describe more complex techniques in the Methods section.* |
| ☐ | ☒ | A description of all covariates tested |
| ☐ | ☒ | A description of any assumptions or corrections, such as tests of normality and adjustment for multiple comparisons |
| ☐ | ☒ | A full description of the statistical parameters including central tendency (e.g. means) or other basic estimates (e.g. regression coefficient) AND variation (e.g. standard deviation) or associated estimates of uncertainty (e.g. confidence intervals) |
| ☐ | ☒ | For null hypothesis testing, the test statistic (e.g. $F$, $t$, $r$) with confidence intervals, effect sizes, degrees of freedom and $P$ value noted<br>*Give P values as exact values whenever suitable.* |
| ☒ | ☐ | For Bayesian analysis, information on the choice of priors and Markov chain Monte Carlo settings |
| ☐ | ☒ | For hierarchical and complex designs, identification of the appropriate level for tests and full reporting of outcomes |
| ☐ | ☒ | Estimates of effect sizes (e.g. Cohen's *d*, Pearson's *r*), indicating how they were calculated |

*Our web collection on statistics for biologists contains articles on many of the points above.*

## Software and code

Policy information about availability of computer code

| Data collection | No software or code was used to collect data. |
|---|---|
| Data analysis | The following statistical software packages were used: Rstudio v.2023.06.1 (package TWANG v.2.6), SPSS v. 28 (Hayes 2022 PROCESS macro v.4.2). Analysis code is available from: https://osf.io/u74bf/?view_only=d27b1d321455470d8dab9a5ae0932029. |

For manuscripts utilizing custom algorithms or software that are central to the research but not yet described in published literature, software must be made available to editors and reviewers. We strongly encourage code deposition in a community repository (e.g. GitHub). See the Nature Portfolio guidelines for submitting code & software for further information.

## Data

Policy information about availability of data

All manuscripts must include a data availability statement. This statement should provide the following information, where applicable:
- Accession codes, unique identifiers, or web links for publicly available datasets
- A description of any restrictions on data availability
- For clinical datasets or third party data, please ensure that the statement adheres to our policy

For S1-S2, prison data is controlled by HMPPS and requests to obtain it should be directed to HMPPS following National Research Committee (NRC) approvals, https://apply-for-hmpps-research.service.justice.gov.uk/Introduction-and-Guidance/. Survey data for this population, which was collected by the researchers, can be obtained from the research team on request.

Data that is not on OSF will be held by the researchers for 10 years as per standard University retention guidelines.
Datasets and output generated for studies 3–9 can be found on OSF at https://osf.io/u74bf/?view_only=d27b1d321455470d8dab9a5ae0932029.

# Research involving human participants, their data, or biological material

Policy information about studies with <u>human participants or human data</u>. See also policy information about <u>sex, gender (identity/presentation), and sexual orientation</u> and <u>race, ethnicity and racism</u>.

| | |
|---|---|
| Reporting on sex and gender | Findings from S1-2 apply to a male population only, which is explained in the SI. The use of gender as a covariate in S3-9 is justified based on established evidence for gender differences in the perception of persons involved in the criminal justice system. |
| Reporting on race, ethnicity, or other socially relevant groupings | We report the distribution of samples' ethnic background composition in all studies.<br>We include ethnicity as a matching criteria & covariate in Study 1, which is necessary to accurately conduct the comparison group analyses. |
| Population characteristics | Studies 1-3 include vulnerable groups (prisoners, ex-prisoners). Studies 3-9 include members of the general population. For sample characteristics, see study design reporting. |
| Recruitment | Samples for S1-2 were recruited from the UK prison population. Samples for S3-9 were recruited from the crowdsourcing platform Prolific. Details on recruitment and reimbursement are provided in the manuscript. |
| Ethics oversight | S1-2 were approved by the ethics board of the University of Oxford and the National Reseach Committee. S3-9 were approved by the ethics board of the School of Anthropology and Connservation at the University of Kent. Details are provided in the manuscript. |

Note that full information on the approval of the study protocol must also be provided in the manuscript.

# Field-specific reporting

Please select the one below that is the best fit for your research. If you are not sure, read the appropriate sections before making your selection.

☐ Life sciences    ☒ Behavioural & social sciences    ☐ Ecological, evolutionary & environmental sciences

For a reference copy of the document with all sections, see nature.com/documents/nr-reporting-summary-flat.pdf

# Behavioural & social sciences study design

All studies must disclose on these points even when the disclosure is negative.

| | |
|---|---|
| Study description | Studies are labelled according to their research design: S1 (quasi-experimental), S2-3, 5-9 (cross-sectional), S4 (experimental) |
| Research sample | Samples for S1 were drawn from the UK prison poulation (provided by the UK Ministry of Justice, from their database P-Nomis), including a group of prisoners who took part in the Twinning Project between 09/2021 and 03/2023 (n = 676, Mage = 31.17, s.d.age = 7.75, White = 59%, Black = 22%, Asian = 7%, Mixed = 9%, Other = 1%, Unknown = 2%) and a control group of prisoners who did not take part. Treatment group (n = 1874, Mage = 34.49, s.d.age = 10.53, White = 73%, Black = 11%, Asian = 8%, Mixed = 5%, Other = 1%, Unknown = 0%).<br>The sample of S2 consisted of people who took part in the Twinning Project in S1 and who took part in a longitudinal survey (n = 388, Mage = 30.20, s.d.age = 7.12, White = 60%, Black = 20%, Asian = 9%, Mixed = 10%, Other = 2%).<br>The samples for S3-9 were recruited via the crowdsourcing platform Prolific, and distinct recruitment are specified.<br>S3, US and UK nationals who have served time in prison (recruited N = 250, final sample N = 249, Mage = 43.88, SDage = 11.63, background US = 66.3%, male (70% male, 27% female, 2% non-binary/third gender, white = 82%, Black = 11%, Asian = 3%, Mixed = 2%, Other = 2%, Unknown = 0%).<br>S4, UK nationals with and without interest in soccer (recruited N = 282, final sample N = 234, 47% soccer fans, Mage = 44.52, SDage 14.05, 65.4% male; 34.2% female; 0.4% non-binary/third gender; 90.2% white, 5.1% Asian, 2.1% Black, 2.1% Mixed, 0.4% Unknown).<br>S5, UK nationals with interest in soccer (recruited N = 331, final sample N = 303; Mage = 41.12, SDage 14.25; 80.5% male; 18.8% female, 0.7% prefer not to say; 82.5% white; 9.2% Asian, 4% Mixed, 1.7% Black, 1% Other, 0.3% North African, 1.3% Unknown).<br>S6, US nationals with an interest in American Football (recruited N = 330, final sample N = 294; Mage = 40.92, SDage 12.47; 96.9% male, 2.4% female, 0.3% non-binary/third gender, 0.3% prefer not to say; 76.2% white, 7.1 % Black, 7.1% Hispanic, 4.1% Mixed, 3.7% Asian, 1.4% Other, 0.3% prefer not to say).<br>S7, US nationals born before 1995 (recruited N = 333, final sample N = 319; Mage = 42.49, SDage 11.69; 57.7% female, 41.4% male, 0.6% non-binary, 0.3% prefer not to say; 74% white,7.8% Black, 6.3% Hispanic, 5% Asian, 3.4% Mixed, 2.5% Other, 0.9% Prefer not to say).<br>S8, UK nationals (recruited N = 330, final sample N = 327; Mage = 42.09, SDage 11.17); 54.7% male, 44.6% female, 0.6% prefer not to say); 92% white, 4.9% Asian, 1.8% Mixed, 0.6% Black, 0.6% prefer not to say).<br>S9, US nationals (recruited N = 330, final sample N = 320, Mage = 44.84, SDage 13.06; 56.6% male, 42.5% female, 0.6% non-binary/third gender, 0.3% prefer not to say; 82.5% white, 6.3% Asian, 5% Black, 3.1% Hispanic, 1.6% Mixed, 1.3% Other, 0.3% Prefer not to say).<br>No samples were representative. |

| | |
|---|---|
| Sampling strategy | Samples for Studies 1 and 2 were recruited based on a pre-determined timeframe, i.e., prisoners who took part in the Twinning Project between 09/2021 and 03/2023. The maximum available data was collected. No statistical methods were used to pre-determine sample sizes, but our sample sizes are larger than those reported in previous publications with comparable study designs (e.g., Kovalsky et al. 2021; McDavid et al., 2019), and larger than samples require for PSM approaches to replicate RCT results (Campbell & Labrecque, 2022). Data for Studies 3 to 9 were convenience samples collected using the online crowdsourcing platform Prolific. Individual samples from the UK and US were recruited between January and May 2023, and participants received on average the equivalent of £9.00 per hour in financial reimbursement for taking part. Sample sizes were determined by a priori power analyses in G*Power (Faul et al., 2009) to detect small-moderate effect sizes in linear regression/ANOVA analyses (i.e., $f2 = 0.08/0.09$, $f = 0.18$) at the standard .05 error probability at .80 power. |
| Data collection | For S1-2 data was collected via a third party (HMPPS). Prison behaviour data which was provided via the MoJ database P-Nomis reflect routine observations and records compiled by prison staff. The longitudinal surveys used for S2 were administered using pen-and-paper questionnaires by prison officers and intervention provider staff who received specialist training for data collection. These surveys were completed in the respective programme facilities during the first and last session of the intervention, and two months after completing the intervention in prisoner's cells. Paper copies of surveys were later digitised for further analsyes by the research team. For studies 3-9, data was collected using online questionnaires, distributed via the crowdsourcing platform Prolific and using the online survey platform Qualtrics. For the experimental study 4, experimental conditions were displayed in randomised order using Qualtric's block-randomisation feature. |
| Timing | Data for studies 1-2 was collected between 09/2021 and 03/2023, and data for studies S3-9 was collected between 01/2023 and 05/2023. |
| Data exclusions | For S1-2 available data of n = 162 cases were excluded from all analyses (67 due to unconfirmed age >18, 93 female cases) and a further 158 cases were excluded from analyses containing case-note variables, due to incomplete data recording. For S3-9, data was removed based on data quality checks (details are reported in the manuscript and SI). The number of exclusions was: S3 = 1, S4 = 48; S5 = 28; S6 = 36; S7 = 14; S8 = 8; S9 = 10. |
| Non-participation | For studies 1-2, details about attrition in the longitudinal studies is provided in the manuscript. For online studies (S3-9) this is not relevant. |
| Randomization | In study 4, participants were randomly presented with stimuli of a within-subject experimental conditions using a block-randomisation function of the online survey software Qualtrics. There were no randomised elements in the assignment to treatment groups or data collection for Studies 1 and 2 which reflect restrictions in the programme delivery and data collection. There was no randomisation in any of the other studies (3, 5-9). Covariates were used to match treatment and control groups using a propensity score weighting approach in S1 (details are reported in the manuscript and SI), and covariates are included (held constant) in all final analyses for S1-9. |

# Reporting for specific materials, systems and methods

We require information from authors about some types of materials, experimental systems and methods used in many studies. Here, indicate whether each material, system or method listed is relevant to your study. If you are not sure if a list item applies to your research, read the appropriate section before selecting a response.

## Materials & experimental systems

| n/a | Involved in the study |
|---|---|
| ☒ | ☐ Antibodies |
| ☒ | ☐ Eukaryotic cell lines |
| ☒ | ☐ Palaeontology and archaeology |
| ☒ | ☐ Animals and other organisms |
| ☒ | ☐ Clinical data |
| ☒ | ☐ Dual use research of concern |
| ☒ | ☐ Plants |

## Methods

| n/a | Involved in the study |
|---|---|
| ☒ | ☐ ChIP-seq |
| ☒ | ☐ Flow cytometry |
| ☒ | ☐ MRI-based neuroimaging |

## Plants

| | |
|---|---|
| Seed stocks | *Report on the source of all seed stocks or other plant material used. If applicable, state the seed stock centre and catalogue number. If plant specimens were collected from the field, describe the collection location, date and sampling procedures.* |
| Novel plant genotypes | *Describe the methods by which all novel plant genotypes were produced. This includes those generated by transgenic approaches, gene editing, chemical/radiation-based mutagenesis and hybridization. For transgenic lines, describe the transformation method, the number of independent lines analyzed and the generation upon which experiments were performed. For gene-edited lines, describe the editor used, the endogenous sequence targeted for editing, the targeting guide RNA sequence (if applicable) and how the editor was applied.* |
| Authentication | *Describe any authentication procedures for each seed stock used or novel genotype generated. Describe any experiments used to assess the effect of a mutation and, where applicable, how potential secondary effects (e.g. second site T-DNA insertions, mosiacism, off-target gene editing) were examined.* |

