## [Peer Review File · Nature Human Behaviour]

Peer Review Information

Journal: Nature Human Behaviour

Manuscript Title: A soccer-based intervention improves incarcerated individuals' behaviour and public acceptance through group bonding

Corresponding author name(s): Martha Newson

Reviewer Comments & Decisions:

Decision Letter, initial version:

6th March 2024

Dear Dr Newson,

Thank you once again for your manuscript, entitled "Social Cohesion May Be Antidote to Global Prison Crisis," and for your patience during the peer review process.

Your manuscript has now been evaluated by 3 reviewers, whose comments are included at the end of this letter. Although the reviewers find your work to be of interest, they also raise some important concerns. We are interested in the possibility of publishing your study in Nature Human Behaviour, but would like to consider your response to these concerns in the form of a revised manuscript before we make a decision on publication.

To guide the scope of the revisions, the editors discuss the referee reports in detail within the team, including with the chief editor, with a view to (1) identifying key priorities that should be addressed in revision and (2) overruling referee requests that are deemed beyond the scope of the current study. We hope that you will find the prioritised set of referee points to be useful when revising your study. Please do not hesitate to get in touch if you would like to discuss these issues further.

In particular, your revision must address the following (as well as all other reviewer comments):

(1) Reviewer 3 raises important questions on ethics-related aspects of your work. Please address these questions in detail, both in your response to the reviewer and in the manuscript itself. To enable reviewers and editors to evaluate this aspect of your work, please provide a copy of your research protocol and all approvals. Please note that we may engage a reviewer with ethics expertise in this

domain in the following round of review.

(2) Reviewers 2 and 3 have suggested improvement in literature review. Please situate the work in the existing literature, such sports' effect on prison population. ensure that you link the current work with up-to-date findings. Relatedly, please thoroughly discuss the findings and the implications. Ensure that you avoid overinterpretations of results.

(3) In addition, all reviewers raise question on methods. Please demonstrate each study with clearer information on participants and study design. Please justify the measure and design validity per Reviewers 2's and 3's requests.

(4) Reviewer 1 requests that you report the results of a power test. However, please only run and report this if you conducted a power test a priori. If this is not the case, please perform and report a power sensitivity analysis. This should demonstrate the power of your statistical test across a range of possible effect sizes that includes the smallest theoretically or practically meaningful effect size. (Please see our related editorial for more information: <https://www.nature.com/articles/s41562-023-01586-w>)

(5) While Reviewer 1 asks for a reordering of the manuscript, please note that our journal formatting requirements ask that your Methods section comes at the end of the main text. Please also note there is not word limit on the Methods section (though we suggest this does not exceed 3000w). Please therefore keep your Methods section at the end of the paper, and include Methods details in full without concern for word limit restrictions. Supplementary Information should be used only for information that is supplementary to the paper (and not necessary to evaluate the strength of the evidence).

In sum, we invite you to revise your manuscript taking into account all reviewer and editor comments. We are committed to providing a fair and constructive peer-review process. Do not hesitate to contact us if there are specific requests from the reviewers that you believe are technically impossible or unlikely to yield a meaningful outcome.

We hope to receive your revised manuscript within two months. I would be grateful if you could contact us as soon as possible if you foresee difficulties with meeting this target resubmission date.

- Include a “Response to the editors and reviewers” document detailing, point-by-point, how you addressed each editor and referee comment. If no action was taken to address a point, you must provide a compelling argument. When formatting this document, please respond to each reviewer comment individually, including the full text of the reviewer comment verbatim followed by your response to the individual point. This response will be used by the editors to evaluate your revision and sent back to the reviewers along with the revised manuscript.
- Highlight all changes made to your manuscript or provide us with a version that tracks changes.

[REDACTED]

We look forward to seeing the revised manuscript and thank you for the opportunity to review your work. Please do not hesitate to contact me if you have any questions or would like to discuss these revisions further.

Sincerely,

[REDACTED]

Reviewer expertise:

Reviewer #1: prison-based intervention, substance use, offending

Reviewer #2: reoffending, social climate, quality of prison life

Reviewer #3: gender relations, restorative justice, violence and recovery

REVIEWER COMMENTS:

Reviewer #1:

Remarks to the Author:

I found this to be an interesting paper examining the impact of a soccer-based program that targets social bonding. The authors present a number of different analyses that examine not only the effects of the program but also the likelihood that individuals in the general population would be willing to employ formerly incarcerated people. An extreme amount of work has been undertaken to analyse the substantial amount of data collected. However, while the different analyses are insightful, the number of sub-studies makes it difficult to go into depth for any of them, with much of the methods information for the studies in supplementary materials. This is to the detriment of the paper.

Introduction

The introduction is comprehensive and could be cut down to make room for more of the methods to be included in text.

Methods

This section should come after the introduction and before the discussion section. The authors demonstrate their skills and knowledge regarding analytic strategy. However basic information regarding the sample (e.g. demographics), representation of control groups and the British and US citizen samples recruited, and the measures included in all surveys should be included in the body of the paper not supplementary materials (i.e. readers should be able to read the whole paper and get all essential information from the paper without going to the supplementary materials). Information on attrition during the project should be included in text (regardless of presentation an ITT analysis). Outcomes of a power test should also be included in the methods section.

Results

The results are comprehensive and adequately presented.

Discussion

The authors have done a good job in the discussion linking findings back to prior research. However, the discussion lacks a thoughtful and engaging section of at least a paragraph that focuses on the implications of their findings on prison policy (e.g. prisoner wellbeing, reintegration, continuum of care), practice (e.g. prison programs) and future research.

The concluding paragraph also needs to be improved and focus on the study itself.

Reviewer #2:

Remarks to the Author:

This is an interesting article on an unusual and promising project. The intervention is theorized as offering social connection. The data collected is high quality and the findings are of interest. I would certainly recommend publication.

Some comments intended to strengthen the article:

I have some reservations about the claim that the research ‘indicates that interventions offering meaningful social connections are a crucial step in alleviating the global prison crisis’. How would it do that? That is, what are the steps required? Not all of the global prisons crisis is about reoffending after release, or communities accepting employees with prison records – so it seems a somewhat exaggerated or naïve claim. I would suggest either being more specific about just how it would contribute, or tone it down slightly.

The authors claim:

we consider the potential for strong forms of group alignment to change this situation by enabling prisoners to bond with law-abiding values and healthier lifestyles while simultaneously fostering the willingness of the public at large to welcome the formerly incarcerated back into the community.

How strong is the evidence that the programme enabled prisoners to ‘bond with law-abiding values and healthier lifestyles’?

I also find the following claims a bit unlikely:

Adjudications refer to offences in prison that require an official hearing and are considered to be an objective measure of behaviour, as well as a relatively good predictor of future reoffending (McDougall et al., 2017).

And

Optimism about desistance from crime is a well-established factor of 103 reoffending in the criminological literature (Kazemian & Maruna, 2009; Villman, 2021), providing this research with a tentative assessment of the extent to which participation in the Twinning project might positively impact reoffending rates.

These are huge leaps in argument, with contested evidence (e.g. adjudications may reflect policing styles on particular wings, optimism can lead to unrealistic but easily disappointed expectations, and so

on) so I wonder whether a better account of the evidence base (and its limitations) might be required?

What are the implications of the positive gardening programme results for any claims made about football per se?

I was surprised not to see any reference to the work of Rosie Meeks on sport and its effects on prison populations.

Reviewer #3:

Remarks to the Author:

This is an interesting and timely manuscript on social cohesion and the global prison crisis. It includes several studies in both the UK and US including multiple studies of a soccer-based intervention program in the UK. However, it was a bit difficult to follow given the nine studies total and the journal's format and some key details are included in the supplemental materials which is quite lengthy and some of it should be noted in the main manuscript. Furthermore, while the prison crisis is global, the studies highlighted in this manuscript are mainly based in the UK with some additional studies in the US. However, the studies on the intervention program (including S1 which uses a quasi-experimental design) are only based in the UK. Thus, the "global" in the title is a bit misleading.

In the introduction, some of the citations are a bit dated. Additionally, there are some typos in the first paragraph. Line 33 "costs the costs" should just be "the costs" and there is an extra space in line 34 before Newton.

For the study on the intervention, it is unclear why a 2 month period was selected to compare indicators of prison behaviour. Additionally, in the results it is noted that demographics were matched; however, there is no mention that the analysis focused only on men. Furthermore, the manuscript would benefit from additional background information being included in the supplemental material on the Twinning Project intervention. It is unclear why the number of sessions ranged from 5-12 and on average how many participants there are in each session.

Besides the studies being approved by university ethics boards, since some of the studies involved prisoners, were additional ethics approvals needed from the prisons themselves? Also, in the supplemental material it is noted that the coaches and prison staff received research training prior to data collection and in the manuscript, it is noted that they distributed the survey. What additional approvals were needed for them to be engaged in the research and how was confidentiality maintained if they distributed the surveys?

The manuscript uses the term "soccer" yet the supplemental materials uses the term "football" and

“American football” and this further adds to the confusion of the manuscript and related materials. A consistent term should be used throughout, or it should be explained especially since the term “American football” does not mean “soccer”.

Conceptually this is a very interesting manuscript. However, as described above, there are some issues that need to be addressed before it can be considered for publication. It is hard to follow and a bit confusing at times and the format required by the journal only makes it more difficult to follow given the methods follow the results and discussion. Additionally, some critical information is missing from the main manuscript.

Author Rebuttal to Initial comments

REVIEWER COMMENTS:

Reviewer #1:

1.1 I found this to be an interesting paper examining the impact of a soccer-based program that targets social bonding. The authors present a number of different analyses that examine not only the effects of the program but also the likelihood that individuals in the general population would be willing to employ formerly incarcerated people. An extreme amount of work has been undertaken to analyse the substantial amount of data collected. However, while the different analyses are insightful, the number of sub-studies makes it difficult to go into depth for any of them, with much of the methods information for the studies in supplementary materials. This is to the detriment of the paper.

We greatly appreciate the reviewer’s recognition of the amount of work undertaken and their helpful comments. We have tried to communicate the sub-studies more effectively to ensure they receive sufficient depth; for instance, with more text given to their methods.

We have now removed any references to parts of the study designs that are not of relevance to the results presented in this manuscript (i.e., the priming of different group contexts), and we have added clarification as it relates to the pre-registration status of S5-9 in the supplementary materials, which reads:

SI E, P47P “The association between identity fusion (to a job applicant) and willingness to hire/ perceived future chances of the applicant was originally pre-registered to be tested as part of a larger model i.e., as a mediator between experimental triggers of fusion and reintegration support for the applicant. The evaluation of the experimental effects on identity fusion and

indirect effects on reintegration support for an ex-prisoner are reported as part of another manuscript (REDACTED, under review). The results reported here do not change when tested as part of the original model. Any overlap in reporting of results has been/will be declared”

1.2 Introduction

The introduction is comprehensive and could be cut down to make room for more of the methods to be included in text.

Fortunately, the editor has granted a suggested maximum of 2,000 additional words for the Methods, which we have now used. In view of this and given that the other two reviewers emphasized the need to situate the present research in further literature, we decided not to cut down the introduction.

1.3 Methods

This section should come after the introduction and before the discussion section. The authors demonstrate their skills and knowledge regarding analytic strategy. However basic information regarding the sample (e.g. demographics), representation of control groups and the British and US citizen samples recruited, and the measures included in all surveys should be included in the body of the paper not supplementary materials (i.e. readers should be able to read the whole paper and get all essential information from the paper without going to the supplementary materials). Information on attrition during the project should be included in text (regardless of presentation an ITT analysis). Outcomes of a power test should also be included in the methods section.

We have respectfully kept the Methods section at the end of the paper, reflecting Nature Human Behaviour’s stipulated formatting. However, we have made improvements to the information shared by including further information.

In the *Data collection* section, we now report all sample sizes and demographic breakdowns (including the control group of S1), and power tests. For all cross-sectional studies, we report the final sample sizes in the manuscript and refer to the supplementary information for details on recruitment processes and exclusion criteria.

In the *Materials* section, we now list each measure that is referred to in the manuscript and provide references to the relevant sources where applicable. We further provide a sample item (unless it’s a self-explanatory single-item measure, e.g., age) and the anchors for the measurement scale. For multi-item measures, we indicate the number of items and the internal reliability score (Cronbach’s alpha), or the average internal reliability score for measures that are used in multiple studies.

PP 18-20. “Study 1 materials reflect the data typically captured within British prisons, shared with us by HMPS. The key outcome variables were observational recordings of adjudications, self-harm incidents and positive/negative case notes, as well as demographics (age, ethnicity) and criminal history variables (Copas score, Copas & Marshall, 1998; index offence; prison behaviour assessment - Incentives and Earned Privileges (IEP) - levels).

Study 2 materials reflect part of an original longitudinal pen-and-paper survey distributed among a subsample of Twinning Project participants at the beginning, end, and two months after the programme, distributed by prison and soccer club staff involved in the programme. Survey items were answered using a 1 = strongly disagree – 5 = strongly agree Likert-scale format unless specified otherwise. Survey measures captured, among other things, participants’ social bonding (social identification, Postmes et al., 2013, “I identify with [x]”; pictorial identity fusion, Swann et al., 2009, “Overlap between circles of Self and Group” A = no relationship – E = total oneness) with reference to different groups (Twinning Project, family, soccer fans, country, a criminal group) and future optimism about employment and desistance from crime (“What do you think your chances are to have/keep a good job?/ stay out of trouble with the law?”, 1 = poor – 5 = excellent) .

Studies 3 – 9 were online surveys conducted with Qualtrics. Study 3 contained measures on future optimism, social bonding (pictorial identity fusion) to different groups (e.g., family, close friends, a criminal group, my country), as well as time spent (days per week) and values associated with the respective groups (honest, law-abiding, supportive, 1 = not at all – 4 = a lot) . The survey also measured self-reported criminal behaviour since release from prison (10 items, e.g., traffic offence, theft, assault, 1 = never – 5 = often, Cronbach’s $\alpha = .84$), and pro-criminal attitudes (Shields & Simourd, 1991, 9 items e.g., “Successful people break the law to get ahead”, 1 = disagree – 3 = agree, Cronbach’s $\alpha = .80$). The study contained additional control variables, including demographics (age, ethnicity, education, nationality, relationship status, parenthood status, employment status), social desirability (Stöber, 2001, 16 items e.g., “I always admit my mistakes open and face the potential negative consequences”, 1 = true – 0 = false, mean Cronbach’s $\alpha = .76$), and subjective socio-economic status (Adler et al., 2000, “Where would you put yourself on the ladder?” 1 = Worst off – 10 = Best off) .

In Study 4, each participant (in the role of a hiring manager) was presented with three descriptions (in random order) of male job applicants who served comparable sentences in

prison and who either successfully completed the Twinning Project, a gardening programme or no educational programme (reflecting the levels of an experimental condition). The two educational programmes were described with regard to their third-party certification, duration and basic learning outcomes. Participants completed measures of perceived transferrable skills of the applicant (8 items, “To what extent does the applicant possess the following skill? E.g., problem-solving”, 1 = not at all – 4 = a lot, mean Cronbach’s α = 87.7), soccer fandom (“Do you consider yourself a football fan”, 1 = yes, 0 = no), willingness to hire (1 = definitely unwilling – 6 = definitely willing), and the perceived future chances of the applicant (“How would you rate [X]’s chances to have/keep a good job?/ stay out of trouble with the law?”, 1 = poor – 5 = excellent). The study also included a combination of control variables relevant for attitudes towards offenders (open-mindedness Levy et al., 1998, 8 items e.g., “The kind of person someone is, is something very basic about them and it can’t be changed very much.”, 1 = strongly disagree – 6 = strongly agree, mean Cronbach’s α = .96), political ideology (Klingemann, 1998, 3 items e.g., “How would you place your views on this scale when you think about social issues?”, 0 = left – 10 = right, mean Cronbach’s α = .95), contact with ex-prisoners (Hirschfield & Piquero, 2010, “How many people do you know personally or professionally who have been to prison?, 1 = none, 4 = many), criminal victim status (“Have you or a family member ever been a victim of crime?, 1 = Yes, 0 = No), hiring experience (“How many hiring decisions do you estimate you have been involved in?” 1 = None, 7 = more than 50), subjective socioeconomic status ,and level of education.

Studies 5 – 9 measured participants’ social bonding with a job applicant (verbal identity fusion, Gomez et al., 2011, 4 items e.g., “I have a deep emotional bond with the applicant, 1 = strongly disagree, 7 = strongly agree, mean Cronbach’s α = 87.2), their willingness to hire, perceived future chances of the applicant (to desist from crime/ to have a good job), and included the same control variables described for Study 4 (S8-9 did not measure social desirability & perceived future chances). We list all measures and respective items (incl. internal reliability scores for multi-item measures), as well as additional contextual information provided to participants for all studies in Supplementary Information A – I.”

1.4 Results

The results are comprehensive and adequately presented.

Thank you.

1.5 Discussion

The authors have done a good job in the discussion linking findings back to prior research. However, the

discussion lacks a thoughtful and engaging section of at least a paragraph that focuses on the implications of their findings on prison policy (e.g. prisoner wellbeing, reintegration, continuum of care), practice (e.g. prison programs) and future research.

The concluding paragraph also needs to be improved and focus on the study itself.

We have added additional text to the discussion regarding implications and future research, as follows:

P8 “Our findings have important implications for prison policy, practice, and future research. The positive impact of the Twinning Project on adjudications compared to a control group underscores the significance of social bonding in shaping prisoner behaviour and attitudes. This emphasizes the need to invest in interventions that foster group identities among incarcerated individuals, as good prison behaviour makes the prison estate a more positive place to live and work for both inmates and staff (Auty & Liebling, 2020). Meek and colleagues paved the way for academic research into prison service sports programmes and the potential to turn prisoners’ lives around. For instance, sports interventions provide an opportunity for physical activity, with associated positive effects on dopamine, mood regulation, sustained concentration and a host of physical and mental wellbeing factors likely to play into desistance behaviours (Meek & Ramsbotham, 2013; Meek & Lewis, 2014). Additionally, sports can provide a potentially unparalleled locus for much needed social connections, as our evidence on the effects of the Twinning Project clearly demonstrates.

Additionally, with a proven effect on adjudications, our data also suggest that there may be potential for economic savings associated with such programmes (i.e., less solitary confinement and fewer adjudication hearings that may in turn contribute to shorter sentences). We also predict that the present findings will relate to reduced reoffending rates for Twinning Project participants, data which will be analysed after the current cohort has been released. However, while social identification is a critical first step, the study suggests that further investment in programs allowing participants to reap the benefits of emergent identity fusion for positive long-term behavioural outcomes. Moreover, the study highlights the complex role of group identities, indicating that while fusion to positive group identities like the nation correlates with better behaviours, fusion to criminal groups undermines law-abiding attitudes. This reflects the need for policy to support interventions that promote positive societal norms and values, facilitating successful reintegration. Precisely who determines these norms is a matter of debate and needs careful planning, ideally consulting with those directly affected by crime, including victims, perpetrators, and their families) (Van Ness et al., 2022). More research is needed to understand the cultural nuances that shape attitudes toward formerly incarcerated individuals both within and beyond the contexts studied here.”

We have also improved the final paragraph, which now focuses more on the study itself:

P9: “The allure of major sports clubs and brands to solve global crises lies not only in the billions of dollars in revenue they may contribute to social issues, but in their billions of loyal fans. Our data suggests that soccer fandom offers a powerful pathway to more prosocial, law-abiding identities for people in prison, but that a broad range of educational interventions are needed to help tackle prison stigma among receiving communities. Our findings suggest that interventions combining a strong social element and educational appeal for the public are urgently needed to help address the global prison crisis.”

Reviewer #2:

2.1 This is an interesting article on an unusual and promising project. The intervention is theorized as offering social connection. The data collected is high quality and the findings are of interest. I would certainly recommend publication.

We thank the reviewer warmly for these positive comments.

Some comments intended to strengthen the article:

1.7 I have some reservations about the claim that the research ‘indicates that interventions offering meaningful social connections are a crucial step in alleviating the global prison crisis’. How would it do that? That is, what are the steps required? Not all of the global prisons crisis is about reoffending after release, or communities accepting employees with prison records – so it seems a somewhat exaggerated or naïve claim. I would suggest either being more specific about just how it would contribute, or tone it down slightly.

We have now toned down this claim as follows:

P1 “To help address the global prison crisis, interventions must focus on improved behaviour plus both employment and positive group alignments on release, thus reducing costs of incarceration, while improving opportunities to integrate.”

2.2 The authors claim:

we consider the potential for strong forms of group alignment to change this situation by enabling prisoners to bond with law-abiding values and healthier lifestyles while simultaneously fostering the

willingness of the public at large to welcome the formerly incarcerated back into the community.

How strong is the evidence that the programme enabled prisoners to ‘bond with law-abiding values and healthier lifestyles’?

We agree that the article does not currently demonstrate this sufficiently so have edited the sentence as follows. For reference, in our pilot study, we included scales about what values participants felt the Twinning Project represented, law-abiding values and healthier lifestyles were at ceiling level with no variation whatsoever, meaning that we did not include these scales in subsequent designs.

P1 “we consider the potential for strong forms of group alignment to change this situation by enabling prisoners to bond with law-abiding groups that represent healthier lifestyles. Such interventions can also, in the right circumstances, encourage the general public to welcome the formerly incarcerated back into the community by providing a platform through which employers are able to bond with ex-prisoners”

2.3 I also find the following claims a bit unlikely:

Adjudications refer to offences in prison that require an official hearing and are considered to be an objective measure of behaviour, as well as a relatively good predictor of future reoffending (McDougall et al., 2017).

And

Optimism about desistance from crime is a well-established factor of 103 reoffending in the criminological literature (Kazemian & Maruna, 2009; Villman, 2021), providing this research with a tentative assessment of the extent to which participation in the Twinning project might positively impact reoffending rates.

These are huge leaps in argument, with contested evidence (e.g. adjudications may reflect policing styles on particular wings, optimism can lead to unrealistic but easily disappointed expectations, and so on) so I wonder whether a better account of the evidence base (and its limitations) might be required?

We have moderated our claims and have removed mention of adjudications being relatively 'objective'. Instead, we include additional critiques, as follows:

P2 "Adjudications refer to alleged disciplinary offences in prison that require an official hearing, typically capturing misconduct that is deemed dangerous to other inmates or staff or otherwise violates institutional policies (Trulson et al., 2011). The reporting of alleged offences can be contingent on staff-to-inmate ratios or prison policing styles (Bottoms, 1999), which are often viewed by prisoners as unfair and reflecting policing biases by prison staff (Butler & Maruna, 2015). Nevertheless, records of prison misconduct (Cochran et al., 2014; Heil et al., 2009), and adjudications in UK prisons, have proven to be relatively reliable predictors of future reoffending (McDougall et al., 2017) when triangulated with other evidence."

P3 "Optimism about desistance from crime is a well-established correlate of reoffending in the criminological literature (Kazemian & Maruna, 2009; Villman, 2021). Although we hope that increased optimism from participation in the Twinning project might positively impact reoffending rates, we are also aware that this cohort may be at a greater risk of disappointment, which could have knock-on effects for successful desistance from crime."

2.4 What are the implications of the positive gardening programme results for any claims made about football per se?

The gardening programme was initially selected given that horticultural programmes are widely available and (as per Prisoners' Education Trust, 2021) have comparable characteristics to the football programme (e.g., they are third-party certified and contain an outdoor component). The findings that gardening alumni were rated more positive than those without an educational programme, and as positive as Twinning Project (TP) alumni (regardless of someone's status as a football fan) have two implications. As stated in the results, football fandom was not in itself sufficient to boost the perceived value of TP. Simultaneously, football- and gardening-based programmes were considered similarly valuable among those without interest in football, despite the fact that the latter is a much more established and common vocational profession. This could reflect the popularity and status of football in the UK, and it is plausible that Twinning Project (and football per se) does not have the same appeal in a different national context.

Prisoners' Education Trust. (2021). Course prospectus 2021-2022 – Start a new chapter in your life. Accessed March 14, 2024 <https://prisonerseducation.org.uk/wp-content/uploads/2021/06/Prisoners-Education-Trust-Course-Prospectus-2021-2022.pdf>

This is now more clearly referred to in the results and discussion:

P6 “suggesting that sports identity was not a powerful motivator of reintegration acceptance in its own right and that participants valued the two educational programmes similarly”

P9 “Our data suggests that soccer fandom offers a powerful pathway to prosocial identities for people in prison, but that educational interventions of all types help tackle prison stigma among receiving communities. Ultimately, the findings suggest that interventions combining a strong social element and educational appeal for the public hold promise in addressing the global prison crisis and reducing the stigma associated with incarceration, offering a pathway to prosocial identities for individuals in prison and promoting community acceptance upon release.”

2.5 I was surprised not to see any reference to the work of Rosie Meeks on sport and its effects on prison populations.

Thank you, we agree that Meeks' work is highly relevant and it is now cited as follows:

P2: Such programmes could include anything from running (Campana et al., 2023) to football (Grundy & Meek, 2021).

P8: Meek and colleagues paved the way for academic research into prison service sports programmes and the potential to turn prisoners' lives around. For instance, sports interventions provide an opportunity for physical activity, with associated positive effects on dopamine, mood regulation, sustained concentration and a host of physical and mental wellbeing factors likely to play into desistance behaviours (Meek & Ramsbotham, 2013; Meek & Lewis, 2014).

Reviewer #3:

3.1 This is an interesting and timely manuscript on social cohesion and the global prison crisis. It includes several studies in both the UK and US including multiple studies of a soccer-based intervention program in the UK. However, it was a bit difficult to follow given the nine studies total and the journal's format and some key details are included in the supplemental materials which is quite lengthy and some of it should be noted in the main manuscript. Furthermore, while the prison crisis is global, the studies highlighted in this manuscript are mainly based in the UK with some additional studies in the US. However, the studies on the intervention program (including S1 which uses a quasi-experimental design) are only based in the UK. Thus, the "global" in the title is a bit misleading.

We thank the reviewer for their comments and have tried to make the article easier to follow by integrating more of the supplementary material into the manuscript's methods. We agree that our study is limited to the global North (and more specifically the UK) but, with the international reach of soccer, believe that this is a tentative first step toward tackling the global crisis, as social cohesion is integral to all human cultures; we have tried to avoid over-stating the results here by using 'may' in the title.

3.2 In the introduction, some of the citations are a bit dated. Additionally, there are some typos in the first paragraph. Line 33 "costs the costs" should just be "the costs" and there is an extra space in line 34 before Newton.

Apologies for these errors, which have been corrected. Regarding citations, some of the oldest citations pertain to the origins of the social identification theory we utilise (e.g., Abrams & Hogg, 1990; Tajfel & Turner, 1979), which are important to retain for two reasons: (a) to situate the research theoretically and (b) to give credit to the original authors. However, we have updated citations throughout the rest of the introduction. For instance:

P1: If the purpose of prison is to reduce crime by serving as a deterrent and pathway to reform, it would seem to be ineffective for most countries in the world (Auty & Liebling, 2020; Petrich et al., 2021; United Nations, 2023).

P2: Sport may be the ideal platform to capture this reciprocal dynamic. However, a recent meta-analysis suggested that while sports programs in prison have a significant moderate effect on behaviour and a large significant effect on self-esteem and wellbeing, sounder evaluative designs are required to understand more fully why this is the case (Jugl et al., 2023), with high-profile calls for more thorough investigations into the processes by which sports interventions work (Murray et al., 2024). Such programmes could include anything from running (Campana et al., 2023) to football (Grundy & Meek, 2021).

3.3 For the study on the intervention, it is unclear why a 2 month period was selected to compare indicators of prison behaviour. Additionally, in the results it is noted that demographics were matched; however, there is no mention that the analysis focused only on men. Furthermore, the manuscript would benefit from additional background information being included in the supplemental material on the Twinning Project intervention. It is unclear why the number of sessions ranged from 5-12 and on average how many participants there are in each session.

First, we selected a 2-month period before and after the intervention for several reasons:

- This was a reasonable window following the programme during which it was unlikely that participants would leave prison or be transferred to another prison. We had to hit a sweet spot between having enough time to observe the investigated behaviours (i.e., longer periods of observation are better) and short enough that prisoners had not left prison or transferred (i.e., shorter periods). In our pilot, we originally had a 3-month window but found that attrition became problematic, as all participants were intended to have less than 12 months left to serve.
- Prisoners could be held in a single prison during the programme for the purposes of the intervention, but could not be held in a prison following the intervention for the purposes of research – this would not have been ethical. Where prisoners moved prisons, we followed up with surveys in the new prison wherever possible. If prisoners left prison entirely then we were not able to follow up as we did not have their contact details.

We have now made it clearer that the Twinning Project analysis focussed on men, as follows:

P17 “For these analyses, we worked exclusively with the male population, due to the unique needs of the women’s population and highly unbalanced sample sizes (women make up around 5% of the prison population).”

We have included additional information in the Supplementary Information about the programme, both regarding the typical programme design (based on the Template for Intervention Description and Replication, TiDierR; Hoffman et al., 2014) and the average characteristics of cohorts involved in our research. This includes the number of weeks the programme typically ranged from 1-12 weeks (minimum 5 sessions), which was dependent on both prison and club logistics. The number of participants varied according to site, but our understanding was that sites aimed to include 10-18 participants per cohort, which was deemed to be the maximum a coach could effectively work with while proving value for money (i.e., too few participants would be costly to run).

Hoffmann, T. C., Glasziou, P. P., Boutron, I., Milne, R., Perera, R., Moher, D., ... & Michie, S. (2014). Better reporting of interventions: template for intervention description and replication (TIDieR) checklist and guide. *Bmj*, 348.

We now include the following in the ms:

P17: The average initial cohort size was 13.06 participants (SD = 3.80; Range 6 – 24) and the average programme length was 6.07 weeks (SD = 3.31, Range = 1 – 19).

3.4 Besides the studies being approved by university ethics boards, since some of the studies involved prisoners, were additional ethics approvals needed from the prisons themselves? Also, in the supplemental material it is noted that the coaches and prison staff received research training prior to data collection and in the manuscript, it is noted that they distributed the survey. What additional approvals were needed for them to be engaged in the research and how was confidentiality maintained if they distributed the surveys?

Yes, we received ethical approval from the National Research Committee to conduct the prison research (National Research Committee (2019-215), and this has now been stated in the methods alongside University approval. Its omission was an oversight. To protect participant data we also had a DPIA in place with HMPPS.

We did not gain ethical approvals from individual prisoners because it is our understanding that approvals should come from ethics boards. Rather, for all participants who provided data directly to us (e.g., via surveys), we shared information in advance of the research (1-7 days in advance), provided an opportunity to ask questions, then provided a consent form to obtain informed consent. Details about the right to withdraw, data security and many other matters were included in the information sheets. Prison participants received a hard copy of the information sheet to keep. For prisoners who did not complete surveys with us, but whose data we used (i.e., a proportion of the treatment group and all cases in the control group), we did not gain informed consent. For these participants, we adhered to GDPR under a DPIA with His Majesty's Prison and Probation Service. Specifically, processing of personal data without explicit consent is covered under the conditions set out in paragraphs 6 to 28 of Schedule 1 of the Data Protection Act 2018 wherein: "Research is one of the University's charitable objectives, which it carries out in the public interest. The University processes data under this lawful basis as the processing is necessary for the performance of the task in the public interest. As a result, personal data can be processed without consent."

We worked with prison and football coach staff to distribute the surveys as it was deemed less stressful for participants to work with coaches than for additional researchers to come in and conduct research; the coaches were already in the process of building relationships with participants as part of the intervention itself. Coaches were responsible for introducing the research, seating participants so that their answers were confidential and not seen by other participants, reminding everyone that surveys were anonymous, checking everyone had received the information sheets, handing these out where needed, answering questions and signposting where needed, reading out survey questions, and answering any survey questions. Prison staff were responsible for distributing information sheets 1-7 days in advance of the research, collecting surveys from coaches at the end of the session, making a note of participant numbers along with participant names and prison numbers (i.e., creating linking data), and posting participant surveys back to the University using a secure courier – pen and paper surveys were essential as there is no Internet access for prisoners and it was an unfair use of HMPPS staff time to transcribe answers for us. This approach was deemed appropriate in our ethical approvals from Oxford University and the National Research Committee. All members of staff agreed to participate in the research.

3.5 The manuscript uses the term “soccer” yet the supplemental materials uses the term “football” and “American football” and this further adds to the confusion of the manuscript and related materials. A consistent term should be used throughout, or it should be explained especially since the term “American football” does not mean “soccer”.

Thank you for pointing out this inconsistency. We have now used soccer throughout the ms and SM, except for study materials which are described as they were presented to participants (e.g., we refer to the measure *soccer-fandom*, but the item refers to being a *football fan*). Similarly, we distinguish soccer from American Football throughout the MS and SI.

3.6 Conceptually this is a very interesting manuscript. However, as described above, there are some issues that need to be addressed before it can be considered for publication. It is hard to follow and a bit confusing at times and the format required by the journal only makes it more difficult to follow given the methods follow the results and discussion. Additionally, some critical information is missing from the main manuscript.

We are grateful for the constructive feedback provided by all three reviewers which has helped us to further improve the manuscript in a range of ways, including the clarity of exposition and the provision of missing information.

Decision Letter, first revision:

12th July 2024

Dear Dr. Newson,

Thank you for your patience as we've prepared the guidelines for final submission of your Nature Human Behaviour manuscript, "Social Cohesion May Be Antidote to Global Prison Crisis" (NATHUMBEHAV-23113740A). Please carefully follow the step-by-step instructions provided in the attached file, and add a response in each row of the table to indicate the changes that you have made. Please also address the additional marked-up edits we have proposed within the reporting summary. Ensuring that each point is addressed will help to ensure that your revised manuscript can be swiftly handed over to our production team.

We would hope to receive your revised paper, with all of the requested files and forms within two-three weeks. Please get in contact with us if you anticipate delays.

Nature Human Behaviour offers a Transparent Peer Review option for new original research manuscripts submitted after December 1st, 2019. As part of this initiative, we encourage our authors to support increased transparency into the peer review process by agreeing to have the reviewer comments, author rebuttal letters, and editorial decision letters published as a Supplementary item. When you submit your final files please clearly state in your cover letter whether or not you would like to participate in this initiative. Please note that failure to state your preference will result in delays in accepting your manuscript for publication.

In recognition of the time and expertise our reviewers provide to Nature Human Behaviour's editorial process, we would like to formally acknowledge their contribution to the external peer review of your manuscript entitled "Social Cohesion May Be Antidote to Global Prison Crisis". For those reviewers who give their assent, we will be publishing their names alongside the published article.

Cover suggestions

We welcome submissions of artwork for consideration for our cover. For more information, please see

our guide for cover artwork.

ORCID

Non-corresponding authors do not have to link their ORCIDs but are encouraged to do so. Please note that it will not be possible to add/modify ORCIDs at proof. Thus, please let your co-authors know that if they wish to have their ORCID added to the paper they must follow the procedure described in the following link prior to acceptance: <https://www.springernature.com/gp/researchers/orcid/orcid-for-nature-research>

Nature Human Behaviour has now transitioned to a unified Rights Collection system which will allow our Author Services team to quickly and easily collect the rights and permissions required to publish your work. Approximately 10 days after your paper is formally accepted, you will receive an email in providing you with a link to complete the grant of rights. If your paper is eligible for Open Access, our Author Services team will also be in touch regarding any additional information that may be required to arrange payment for your article.

Please note that *Nature Human Behaviour* is a Transformative Journal (TJ). Authors may publish their research with us through the traditional subscription access route or make their paper immediately open access through payment of an article-processing charge (APC). Authors will not be required to make a final decision about access to their article until it has been accepted. Find out more about Transformative Journals

For information regarding our different publishing models please see our Transformative Journals page.

If you have any questions about costs, Open Access requirements, or our legal forms, please contact ASJournals@springernature.com.

[REDACTED]

Best regards,

[REDACTED]

On behalf of

[REDACTED]

Reviewer #1:

Remarks to the Author:

I commend the authors for substantially improving the quality of the manuscript. They have adequately addressed my comments.

Reviewer #2:

Remarks to the Author:

This is a very careful response to of all the reviewer comments received. Happy to approve publication.

Reviewer #3:

Remarks to the Author:

This manuscript is much improved from the previous one. The authors have addressed the issues raised by the reviewers.

Final Decision Letter:

Dear Dr Newson,

We are pleased to inform you that your Article "A soccer-based intervention improves incarcerated individuals' behaviour and public acceptance through group bonding", has now been accepted for publication in *Nature Human Behaviour*.

Please note that *Nature Human Behaviour* is a Transformative Journal (TJ). Authors may publish their research with us through the traditional subscription access route or make their paper immediately open access through payment of an article-processing charge (APC). Authors will not be required to make a final decision about access to their article until it has been accepted. Find out more about Transformative Journals

An online order form for reprints of your paper is available at <https://www.nature.com/reprints/author-reprints.html>. All co-authors, authors' institutions and authors' funding agencies can order reprints using

the form appropriate to their geographical region.

With best regards,

[REDACTED]